# Effects of Digitalized Front-of-Package Food Labels on Healthy Food-Related Behavior: A Systematic Review

**DOI:** 10.3390/bs12100363

**Published:** 2022-09-27

**Authors:** Nikola Ljusic, Asle Fagerstrøm, Sanchit Pawar, Erik Arntzen

**Affiliations:** 1School of Economics, Innovation, and Technology, Kristiania University College, 0152 Oslo, Norway; 2Department of Behavioural Sciences, Oslo Metropolitan University, 0176 Oslo, Norway

**Keywords:** food labeling, consumer behavior, healthy foods, physical labels, digitalized labels, technology

## Abstract

Front-of-package (FOP) food labels may impact healthy food-related behavior. However, such labels may be presented using new technology and they may impact behavior differently than physical labels. This systematic review investigated the effects of physical and digitalized labels on healthy food-related behavior. This review used four search engines to collect articles that investigated the effects of food labels on the purchase, consumption, hypothetical choice, and self-reports of healthy foods. General findings, types of labels, or whether the articles used physical versus digitalized static, interactive, or technology-enabled labels were synthesized. The dependent variables were categorized according to whether they were under full, partial, or no control of the independent variables. The risk of bias was measured by the RoB 2 tool and adapted Joanna Briggs Institute Checklist. The search strategy identified 285 records and 30 articles were included. While digitalized static and physical labels did not differ in their effects on healthy food-related behavior, technology-enabled labels were more predictive of healthy food-related behavior than interactive labels.

## 1. Introduction

Consumption of unhealthy foods is a major societal problem despite numerous efforts by different institutions and organizations. Obesity has approximately doubled worldwide since the 1980s [1]. Research shows a connection between the consumption of unhealthy foods and an increased risk of heart disease [2]. In addition, it is even associated with an increased risk of suicide attempts [3]. Furthermore, it is also an economical burden for society. A high body mass index is estimated to cost USD 990 billion per year globally for healthcare services [4]. Using mandatory nutritional labels on food products is only one of many proposed interventions. It may have ameliorated the rising epidemic of obesity. However, this may not be the case for all subgroups of consumers, such as individuals who are already obese [5]. As a result, the World Health Organization [6] suggests that the food industry should promote healthier diets by providing simple and clear food labels. This can be achieved by presenting simplified front-of-package (FOP) food products. Several types of FOP food labels exist, as shown in Figure 1. However, research shows that the effects of these FOP food labels on healthy food-related behavior are inconsistent and vary in relation to which type of behavior is measured [7,8,9,10,11,12].

Technology may be used to present digitalized FOP food labels in novel ways, and such labels may be more effective than physical labels. Digitalized FOP food labels may be static, interactive, and technology-enabled. Interactive technology may provide detailed product information to consumers, and technology-enabled retailing may provide personalized products for each consumer, dynamic presentations of products that may be changed based on previous purchase history, and provide real-time information where such offers are given immediately [13,14]. These characteristics may be used for digitalized FOP food labels. For instance, Shin et al. [15] presented digitalized FOP food labels in order to study their effect on healthy food purchases. The label presented an overall healthiness score based on foods in the virtual basket of each consumer in an online grocery store experiment. They found that such digitalized FOP food labels increased healthy food purchases. Physical FOP food labels are static labels that are presented on the physical package, menu boards, or shelf tags near the products in physical stores. Digitalized FOP food labels are presented mostly in online grocery stores by a medium or device. In this situation, digitalized FOP food labels are presented together with images of the food product and may be in the form of static, interactive, or technology-enabled. Digitalized static FOP food labels are similar to physical FOP labels as they also present a static image of the food label but differ as they are presented through a medium. Interactive FOP food labels provide additional options to access more information regarding the health aspects of the food product or the label. Technology-enabled FOP food labels provide personalized, dynamic, and real-time information. Specifically, such labels can provide personalized information based on each consumer, dynamical information based on their specific actions with the medium, and real-time information to the consumers. Hence, physical, digitalized static, interactive, and technology-enabled FOP food labels may present different information to consumers, as shown in Figure 2, and these may influence healthy food-related behavior in different ways.

Healthy food-related behavior may be measured in several ways and FOP food labels may impact these behaviors differently. Healthy food-related behaviors can be categorized into purchase, consumption, hypothetical choice, and self-reports regarding healthy foods. Purchase may be measured by actual money spent on foods, consumption in terms of the number of calories consumed, and hypothetical choices may be measured by a relative selection of a product given a set of several products without actually owning or consuming the item in the presence of a question. Self-reports are verbal estimations of participants′ own behavior toward a given product in the context of other questions and may be used to study consumer behavior in general (see [16] for different measurements of consumer behavior related to FOP food labels). FOP food labels may affect one or several of these measurements. For example, the purchase of foods may be influenced by FOP food labels, although with either small or inconclusive results [7], consumption may be influenced by different FOP food label types [10], hypothetical choices may also be impacted by these FOP food labels [17], and self-reports such as participants ratings of healthfulness [18], taste [19], trustworthiness, intent to purchase [20], affect and familiarity [21] related to healthy foods may also be influenced by FOP food labels.

Although there exists extensive research on related topics, few literature reviews exist on the effects of digitalized FOP food labels on healthy food-related behavior. For instance, there exists research on health-related information delivered by technology in physical stores [22] and the effects of health labels and ingredient labels on consumption and self-reports [23]. Furthermore, Granheim et al. [24] did a systematic scoping review regarding the digital food environment and identified some articles which have examined the impact of healthy food labels on healthy food-related behavior in an online grocery setting without comparing their effects to physical FOP food labels. Similarly, Pitts et al. conducted a review on the promises and pitfalls of online grocery shopping related to healthy food purchase [25]. However, to the best of our knowledge, no systematic literature review has examined the effects of physical and digitalized FOP food labels regarding different types of labels and their effect on healthy food-related behavior. Knowledge of such effects on healthy food-related behavior has both academic and social importance. First, such research provides an understanding of how technology in this setting influences human behavior, health-related behavior, and consumer behavior. Second, such knowledge may aid in reducing obesity, economic costs, and human suffering worldwide. Finally, knowledge regarding digitalized FOP food labels may give brand owners and retailers a competitive advantage [26]. Specifically, digitalized interactive and technology-enabled FOP food labels may provide more accurate descriptions regarding products and present personalized, dynamic, and real-time information on health scores to consumers. This may increase the value of healthy foods while at the same time benefiting brand owners and retailers by increasing the number of healthy food purchases, attracting new customers, and increasing positive word-of-mouth [27]. This paper aims to fill that knowledge gap by presenting a classification system of physical and digitalized FOP food labels and investigating how such labels impact consumer behavior through a systematic review. The objective of this paper is to investigate how physical, digitalized static, digitalized interactive, and digitalized technology-enabled FOP food labels impact purchase, consumption, hypothetical choice, and self-reports regarding healthy foods.

The rest of the paper is structured in the following manner. First, previous research regarding the classification of FOP food labels and studies that have investigated physical labels, and digitalized static, interactive, and technology-enabled labels is presented. This is followed by providing the methods used in this review. The result of the review is then presented. Discussion of the results in light of previous research is then provided. At last, further research directions are suggested.

## 2. Literature Review

FOP food labels can be classified as summary labels, nutrient-specific labels, or combinations of both [9] as shown in Figure 1, and their effects may be moderated by other variables. Summary labels present an overall health evaluation of a food product and may be presented as single or graded summary labels. Single summary labels are binary and their presence on a food product indicates that the product is considered healthy; an example is the Nordic Keyhole [28]. However, graded summary labels present a score between a minimum and a maximum value as a higher score corresponds to a higher degree of the healthiness of a product, such as the French Nutri-Score [29]. In contrast, nutrient-specific labels present some key nutrients on the front of the package and specify the degree of the healthiness of specific nutrient contents. Nutrient-specific labels can be presented in terms of percentage-based, single, and graded nutrient-specific labels. Percentage-based nutrient-specific labels show a percentage that is based on specific nutrient content or recommended daily intake based on an average adult. Single nutrient-specific labels are binary and show an excess of a given nutrient. Graded nutrient-specific labels show the nutritional content such as “low”, “medium”, or “high” amounts. Guideline daily amounts [30] warning labels [31], and traffic lights [30] are examples of nutrient-specific labels. Some FOP food labels use combinations of summary and nutrient-specific label elements such as the Australian Health Star Rating system [32]. In addition, several other independent variables that influence the effectiveness of these labels have been investigated, such as color-based labels [8,12], time-pressure conditions, nutritional knowledge about labels [11], textual claims [33], and self-control [34], among others. In short, FOP food labels can be categorized in summary, nutrient-specific, and combined labels with several different subcategories and other variables in combination with labels have been investigated.

Physical FOP food labels and their effects on healthy food-related behavior have been examined by other researchers. Roberto et al. [19] allocated participants randomly in a campus store context to no FOP food label, combined FOP food label, and combined FOP food label with additional per serving information conditions. These labels were presented for a cereal and the study measured participants’ self-reports regarding estimations of calories, total sugars, vitamins, healthfulness, intent to purchase cereals, total grams poured, and total grams consumed. Participants allocated to the combined FOP food label per serving condition had higher self-reports regarding estimations of calories. In contrast, other healthy food-related behavior measurements did not differ between the conditions, indicating no effect. Similarly, Julia et al. [35] allocated participants in a controlled lab store context to no FOP food label, graded summary FOP food label, and graded summary FOP food label with information regarding the criteria of the labels, and these were presented for different food products. The study measured the mean nutritional qualities of the food items participants had selected in their shopping carts and used self-reports regarding the recall of the labels, healthfulness, and understanding. The results show that there were few differences between healthy food-related behavior as a function of these FOP food labels on hypothetical choice, but that they impacted self-reports regarding recall and understanding. Koeningstrofer et al. [34] conducted two studies regarding the effects of nutrient-specific FOP food labels on healthy food-related behavior. In the first study, participants in a controlled laboratory store context were allocated to no FOP food label. Participants in the nutrient-specific FOP food label conditions were instructed to shop for four items. The process measured participants’ purchases and self-reports regarding self-control and used professional dieticians to classify which foods were considered healthy. The second study extended the previous study by using a standardized healthiness of food product scale instead of ratings of dieticians. The results of both studies show that participants who had lower self-reports regarding self-control were correlated with a larger decrease in unhealthy food purchases when the label was present, while participants with higher self-reports regarding self-control were correlated with smaller effects when the label was present. Hence, in regard to the effects of physical FOP food labels, one article found differences regarding purchases when self-reports regarding self-control were high [34] one article did not find different effects on hypothetical choices [35], one article did not find differences in self-reports [19], and one article found different effects regarding self-reports [35].

Digitalized static FOP food labels and their effects on healthy food-related behavior may differ from physical FOP food labels. For instance, digitalized static FOP food labels may be presented at several locations in the online grocery retail setting, such as on the first webpage in context with other products or on the second webpage when the product is presented alone and together with the nutritional information labels of that product. Digitalized static FOP food labels may be enlarged and may take up more space with the food product image than physical labels. In addition, there is a longer delay between the purchase and consumption of food products in the presence of digitalized FOP food labels compared to physical FOP food labels. Talati et al. [17] conducted a large-scale online experiment with participants across 12 countries to study the effects of combined, graded nutrient-specific, percentage-based nutrient-specific, graded summary, single nutrient-specific, and no FOP food labels on hypothetical choice. Participants were exposed to food products with no FOP food label, were instructed to select which one out of three products they would like to purchase, and were again presented with the same products in combination with one type of FOP food label. Their results show that the hypothetical choice regarding healthy foods was improved from most to least by graded summary, graded nutrient-specific, single nutrient-specific, combined, and percentage-based nutrient-specific FOP food labels. Similarly, Raats et al. [36] conducted an online experiment with participants from 6 different countries and investigated the effects of percentage-based nutrient-specific labels based on “per 100g” and “typical portion size” in combination with different food products on self-reports regarding the healthfulness of products by categorizing products on a scale from most to least healthy. Their results show that these labels produced different results. For instance, products in combination with labels based on “per 100g” were rated less healthy than “typical portion size” labels. At last, Khandpur et al. [37] investigated the effects of single nutrient-specific and graded nutrient-specific labels on hypothetical choices and self-reports regarding the intention to purchase, nutritional accuracy, and ratings of the healthfulness of food products. Their results show that participants exposed to single nutrient-specific labels had a higher hypothetical choice and self-reports regarding nutritional accuracy, and lower self-reports regarding healthfulness than graded nutrient-specific FOP food labels. Hence, in regard to the effects of digitalized static FOP food labels, two article found differences in hypothetical choice [17,37], and three articles found differences in self-reports regarding the healthfulness of products [36,37].

Digitalized interactive FOP food labels and their effects on healthy food-related behavior have been investigated, although less than digitalized static FOP food labels, and they may also impact healthy food-related behavior differently than other labels do. For instance, consumers could get more information about the product’s nutritional information or how such labels grade a given food product. In addition, the location of options such as a button on the first screen or the second screen could also influence healthy food-related behavior. Furthermore, there exists research that has examined the effects of digitalized interactive FOP food labels. Egnell et al. [38] investigated the effects of graded summary, percentage-based nutrient-specific, and no FOP food labels on hypothetical choices in an online grocery context. Interestingly, participants in that study had the option to access more information regarding the labels or the food product by clicking a specific button. Participants’ hypothetical choice regarding healthy foods was higher when exposed to graded summary labels than to percentage-based nutrient-specific labels. No label produced the least hypothetical choice regarding healthy foods. Maubach et al. [39] investigated the effects of graded summary, graded nutrient-specific, percentage-based nutrient-specific, and no FOP food labels on best-worst scaling in a choice experiment. Similarly, participants could get more information regarding nutrients, ingredient lists, and allergens by clicking on a specific button. Their results show that graded nutrient-specific labels had the most impact on hypothetical choices than other conditions. Andrews et al. [40] examined single summary labels, graded nutrient-specific labels, and no FOP food labels on self-reports regarding the healthfulness of the product and nutrient estimations of food products. The participants could click on a button to see nutritional labels on the back of the products. Their results show that graded nutrient-specific labels generated higher nutrient accuracy than did single summary labels. Single summary labels generated higher self-reports regarding healthfulness than the other conditions. Sacks et al. [41] examined the impact of graded nutrient-specific labels and no FOP food labels on purchase. Likewise, the participants were presented with the FOP food labels and could get more information about the labels or nutritional information by clicking on a specific button. Their results indicate that introducing these FOP food labels did not change overall purchases nor sales of products without “red labels.” Fuchs and colleagues [42] investigated the effects of interactive FOP food labels on purchase and self-reports of healthy foods in a laboratory-based online grocery store. Specifically, they developed a Google Chrome extension that displayed Nutri-Score for product-specific food products. Their result shows that individuals that were exposed to such static labels purchased on average, more healthy food products than did controls. In addition, the effect was stronger for individuals with low food literacy and individuals that were exposed to such labels showed stronger advocacy for introduction of such labels. Hence, one article shows that digitalized interactive FOP food labels did not influence purchase [41] while one article found an increase in purchase [42], two articles show that digitalized interactive FOP food labels influenced hypothetical choices [38,39], and two articles show that digitalized interactive FOP food labels influenced self-reports of healthy foods [40,42].

Digitalized technology-enabled FOP food labels and their effects on healthy food-related behavior have not been investigated in as much detail as physical or other digitalized FOP food labels. For instance, one could arrange a digitalized technology-enabled FOP food label that presents an overall graded summary label as a personalized, dynamic, and real-time based progress bar based on all products within a virtual basket before or during purchase. Such a progress bar may display how healthy a food shopping cart is or how unhealthy a virtual food cart is. As indicated elsewhere [43], such framing may influence food purchases. However, few articles have investigated the effects of digitalized technology-enabled FOP food labels on healthy food-related behavior. Shin et al. [15] investigated the effects of aggregated dynamic food labels with real-time feedback based on food products in each consumer’s virtual basket, and presented the result as a pie chart based on graded summary FOP food labels or based on graded nutrient-specific FOP food labels, in combination with an option to sort products selected by consumers from most to least healthy on consumers food selection. The study used a crossover design. Half of the participants completed grocery shopping first without the dynamic food label and then with the dynamic food label. The other half completed shopping first with the label and then without the label. The participants who were exposed to the labels could select which one of seven different types of FOP food labels they would like to see. The study results show that participants exposed to the aggregated dynamic real-time food label scores selected on average, foods with a higher Nutri-score value, lower amounts of total sugar, and lower calories than those not exposed to such FOP food labels. Hence, one article [15] showed that digitalized technology-enabled FOP food labels increased healthy food choices.

## 3. Materials and Methods

The procedure for conducting this systematic review was based on the Preferred Reporting Items for Systematic Reviews and Meta-Analyses (PRISMA) statement [44].

### 3.1. Eligibility Criteria

The articles which were included were (a) peer-reviewed journal articles and books, (b) empirical research articles which presented new data, and (c) written in English. With regard to (d) the first screening phase, articles that had the following text in the title, abstract, or keywords: “cues”, front-of-package”, “labels”, “point-of-decision”, “symbols”, “icons”, and “logos” and its effect on food-related behaviors were eligible for the final screening phase. After the search, the following variations of the terms were included in order to avoid ambiguity: “cue”, “label”, “package”, “packaging”, “icon”, and “logo”. Regarding (e) the final screening phase, the articles included in this review were based on the full text of the article and included if they investigated FOP food labels on healthy food-related behaviors. FOP food labels were defined as a single summary, graded summary, percentage-based nutrient-specific labels, single nutrient-specific labels, graded nutrient-specific labels, or combined labels. Healthy food-related behaviors measured participants’ purchases, consumptions, hypothetical choices, and self-reports related to healthy foods measured quantitatively. After the search, self-reports were defined by being assessed on a Likert-type scale. Healthy food was defined as either low in sodium, saturated fats, sugar, or calories, or with an excess of protein, unsaturated fats, fiber, or vitamins. Conference articles, other sources, conceptual articles, literature reviews, articles that used secondary data, non-English articles, and articles that violated the first and final screening criteria were excluded.

### 3.2. Search Strategy

Studies were identified using search engines for academic peer-reviewed journal articles, and the search engines selected were based on the findings by Gusenbauer and Haddaway [45]. The principal search engines that were used for this study were “Web of Science”, “Science Direct”, “PubMed”, and “Wiley Online Library.” The search was performed, and information regarding articles was extracted on the 8 of November 2021. The search consisted of identifying possible eligible articles using the following search string: “front-of-package“ AND (“technology” OR “online grocery”). The same search string was used in all the search engines. In addition, no filters were used during the search in all search engines. There were no imposed restrictions on publication dates or journal categories. The search process consisted of extracting the reference information for possible eligible articles by clicking on the “export” option for each search engine and downloading an article information file. The files contained the following information name of the journal, year of publication of the article, author(s) of the article, title of the article, the abstract, and keywords for each article.

### 3.3. Selection Process

These article information files from the four databases were merged into one common file. Two independent reviewers screened the articles listed in the common article information file based on the eligibility criteria. The reviewers had inter-rater reliability of 85.23% agreement in the first screening phase. The two reviewers resolved disagreements by discussing which eligibility criterion was violated, followed by a reassessment. If the meeting did not result in an agreement, then a third independent reviewer provided a final assessment of whether the article met the eligibility criteria. The reviewers had inter-rater reliability of 72.72% agreement in the final screening phase. Similarly, disagreements for the final screening phase were performed by two independent reviewers, and a final assessment by a third if there were disagreements. The consensus of the two reviewers resolved all disagreements.

### 3.4. Data Collection Process

The data collection process consisted of using a data collection sheet, and collection was performed by one reviewer. Data were obtained by identifying each of the article’s information in the common article information file and based on the full text of the articles. The data of the full-text articles were extracted on 10th December 2021. The data items on the data collection sheet consisted of the year of publication, name of the journal of the article, name of authors, name of title, the abstract, number of observations included in the analysis, unit of analysis, percentage of female participants, research design, controlled or field setting, dependent variable(s), independent variable(s), comparison of data method, effect strength, univariate or multivariate independent variables, findings of the study, type of FOP food labels, physical or digitalized FOP food label, and whether the FOP food labels were static, interactive or enabled by technology. The research designs were categorized into between-participant, within-participant, and non-experimental surveys. The dependent variables were categorized into purchase, consumption, hypothetical choices, and self-reports regarding healthy foods. Self-reports were defined as verbal estimations by participants measured by Likert-type scales. The type of FOP food label was categorized by single summary labels, graded summary labels, percentage-based nutrient-specific, single nutrient-specific labels, and graded nutrient-specific FOP food labels. The findings of the studies were summarized by describing the methods and results of each included study based on the participants, intervention, control condition, and outcome measurement. Physical FOP food labels were defined as labels being presented near the three-dimensional package of a product, digitalized static FOP food labels were defined as being a label presented on a picture of the product, digitalized interactive FOP food labels were defined with the same criteria as digitalized static FOP food labels but with the additional option to view more information of the label or food product, and digitalized FOP food technology-enabled labels were defined as labels that presented information which was personalized, dynamic, and real-time based on participants actions in the study.

### 3.5. Synthesis of Results

Six syntheses of results were used in this review. First, a methodological overview of each article was synthesized in a table by its article number and the first 11 data items (except item 2) specified in the data collection process. Second, FOP food labels used in included articles were synthesized in a table by article number, year of publication, author(s), whether the study used physical or digitalized FOP food labels, whether they were static, interactive or technology-enabled, and which type of FOP food label was used. Third, the findings of each article were synthesized in a table by article number, year of publication, author(s), and findings which were summarized by the method and results by describing the participants, intervention, control condition, and outcome variable (dependent variable) used in each article. Fourth, the number of articles that investigated physical and digitalized FOP food labels as a function of the year of publication is represented by a bar graph. Fifth, articles that investigated the effects of FOP food labels’ presence compared to their absence on the dependent variable were synthesized by presenting how many articles investigated the physical, digitalized, digitalized static, digitalized interactively, and digitalized technology-enabled FOP food labels; and the percentage of articles that found that the dependent variables were under experimental control of these labels. Finally, articles that investigated the effects of FOP food labels’ presence compared to their absence across the dependent variables were synthesized by presenting how many articles investigated the physical, digitalized, digitalized static, digitalized interactively, and digitalized technology-enabled FOP food labels, and whether purchase, consumption, hypothetical choice, or self-reports separately changed as a function of these labels.

### 3.6. Study Risk of Bias Assessment

This study used a risk of bias assessment based on the RoB 2 tool [46] for studies that used randomized controlled trials and an adapted Joanna Briggs Institute (JIB) checklist for non-randomized controlled studies [47] for all studies included in the review. For the randomized controlled trials studies, each study was assessed for bias due to the randomization process, deviation from intended intervention, missing data, measurement of outcomes, and reported results. An adherence assessment was used for deviations from intervention, and the additional risk of bias assessment was given for crossover and cluster randomized controlled trials. For the non-randomized controlled trials, each study was assessed for risk of bias regarding temporal relations between independent variables and their effects, participants’ characteristics across groups, a clear procedure for each intervention, a control condition, multiple measurements of the outcome, missing data, measurement of outcome, and reliability of the outcome. Both the RoB 2 tool and adapted JIB Checklist were used to evaluate an overall risk of bias score, indicated by “Low risk”, “Moderate risk”, or “High risk” for each study. The original assessment for JIB was changed from “Yes”, “No”, or “Unclear” to the assessment mentioned above.

## 4. Results

### 4.1. Study Selection

A visual representation of the study selection process is shown in Figure 3. The search strategy resulted in the identification of 285 records. Fourteen of them were duplicates. They were removed. Two hundred and seventy-one were screened based on the first screening criteria, and 216 records were excluded for not meeting the criteria. All remaining 55 reports were sought for retrieval and assessed for eligibility. Out of those, a total of 25 reports were excluded based on the final eligibility criteria, as 10 of the reports did not investigate FOP food labels as defined in this review, five of the reports did not investigate behaviors related to healthy foods, four of the reports did not investigate the dependent variable specified in this review, and one report did not use primary data collection. This resulted in a total of 30 articles included in this review [15,33,48,49,50,51,52,53,54,55,56,57,58,59,60,61,62,63,64,65,66,67,68,69,70,71,72,73,74,75].

### 4.2. Study Characteristics

The methodological approach for each included article is shown in Appendix A Table A1. The most common units of analysis were participants. The studies had a variability range of 1902 regarding the participants, and approximately two-thirds of the studies had a female participant percentage between 30% and 60%. In regard to the research design of the articles, the majority were between-participant research design (52.9%), followed by within-participant research design (47.1%), and non-experimental surveys (0%). The majority of the studies were conducted in a controlled setting (76.7%). From most to least common approaches for measuring the dependent variable, the articles used self-report (46.7%), hypothetical choice (40%), purchase (11.1%), and consumption (2.2%). The independent variables that were investigated were different types of FOP food labels, labels with different product categories, nutritional information labels, different food products, textual health claims, brands, color-based labels, loss or gain framing, the time limit to shop, caloric information on each ingredient, caloric information relative to other ingredients, amount of labels within a food category, its correspondence to nutritional information, dynamic and real-time feedback, and preparation method for products. The three most common comparisons of data methods were ANOVA, t-tests, and chi-square tests, and the majority of the studies investigated multivariate independent variables.

The FOP food labels from each article are shown in Appendix A, in Table A2. Regarding physical and digitalized FOP food labels, seven articles investigated physical FOP food labels, and 23 articles investigated digitalized FOP food. Out of all included articles, 24 articles investigated static labels, six articles investigated interactively, and one article investigated technology-enabled FOP food labels. Regarding the type of FOP food labels, 24 out of 30 articles investigated nutrient-specific labels, while 12 out of 30 articles investigated summary labels. Specifically, 18 articles investigated graded nutrient-specific labels, 10 investigated single nutrient-specific labels, seven investigated graded summary labels, five investigated single summary labels, and five articles investigated percentage-based nutrient-specific labels.

The findings from each article are shown in Appendix A, in Table A3. Based on the findings of all 30 articles, 18 articles documented different values of the dependent variables in the presence of FOP food labels compared to the absence of such labels. Five articles found different effects of FOP food labels depending on which dependent variable was used. One article found no difference between the presence and absence of FOP food labels. The six remaining articles lacked an absence of label condition. In regard to all articles, 10 articles found differences between different types of FOP food labels, three articles found differences depending on which dependent variable was used, and 17 articles did not compare different types of FOP food labels as categorized by this review (e.g., some of the articles investigated several single nutrient-specific labels) or investigated only one label.

### 4.3. The Effects of Physical and Digitalized FOP Food Labels

The number of articles that have investigated either physical and digitalized FOP food labels and the year of publication is shown in Figure 4. The figure shows that the number of articles investigating digitalized FOP food labels increased steadily from 2011 to 2019 and that digitalized FOP food labels were higher in 2020 and 2021 than were physical FOP food labels.

Articles that investigated the effects of the presence of physical and digitalized FOP food labels compared to their absence are shown in Appendix A, in Table A4. The effects of the presence of physical FOP food labels compared to their absence were investigated by six articles in this review. Out of those, five articles documented that the presence of physical FOP food labels was associated with a change in the dependent variables compared to the absence of FOP food labels. In contrast, the remaining articles had different results depending on the dependent variables being measured. The effects of the presence of digitalized static FOP food labels compared to their absence were investigated by 12 articles. Out of those, ten articles documented that the presence of digitalized static FOP food labels was associated with a change in the dependent variables. Two articles documented mixed results depending on which dependent variable was used. Five articles investigated the effects of digitalized interactive FOP food labels. Three of these documented a change in the dependent variable. One article found mixed results, and one article did not find differences. The effects of digitalized technology-enabled FOP food labels were investigated by one article, and it documented that the FOP food labels did change the dependent variables.

Articles in this review that investigated the effects of the presence of physical and digitalized FOP food labels compared to their absence across the dependent variables are shown in Table 1. In regard to purchasing as the dependent variable, one article did not find differences when exposed to physical FOP food labels, no articles examined digitalized static FOP food labels, two out of three articles found differences when exposed to digitalized interactive FOP food labels, and one article found differences when exposed to digitalized technology-enabled FOP food labels compared to the absence of such labels. Regarding consumption as the dependent variable, one article found differences when participants were exposed to physical FOP food labels compared to their absence, and no articles examined digitalized FOP food labels. Regarding hypothetical choice, two articles found differences when exposed to physical FOP food labels, all eight articles found differences when exposed to digitalized static FOP food labels, one out of two articles found differences when exposed to digitalized interactive FOP food labels, and no articles investigated the effects of digitalized technology-enabled FOP food labels. Regarding self-reports as the dependent variable, two out of three articles found differences when exposed to physical FOP food labels, three out of five articles found differences when exposed to digitalized static FOP food labels, no articles investigated the effects of digitalized interactive FOP food labels and digitalized technology-enabled FOP food labels.

The most effective type of FOP food labels compared to other labels and their impact on purchase, consumption, hypothetical choice, and self-reports regarding healthy foods were also investigated. One article investigated multiple types of FOP food labels regarding physical FOP food labels and found that combined labels were most effective in changing the dependent variable. Specifically, Koeningstrofer et al. [51] investigated the effects of graded nutrient-specific, single summary, and combined FOP food labels on hypothetical choices and found that participants who were exposed to combined FOP food labels selected food products that had the least harmful nutrients based on the SSAg/1 scale [49]. Six articles investigated the effects of multiple types of FOP food labels regarding digitalized static FOP food labels. Three articles identified which type of FOP food label was most effective in changing the dependent variable, while the three remaining articles found inconclusive results. Two articles found graded nutrient-specific labels, and one found that graded summary labels were most effective. Specifically, Gustafson & Zeballos [62] investigated the effects of percentage nutrient-specific and graded nutrient-specific FOP food labels and found that graded nutrient-specific labels were most effective. Hagmann & Siergrist [63] investigated the effects of the graded nutrient-specific and summary labels and found that graded nutrient-specific labels were most effective. Gabor et al. [66] investigated the effects of graded nutrient-specific, graded summary labels, and percentage nutrient-specific labels and found that graded summary labels were the most effective. Deliza et al. [55] investigated percentage nutrient-specific labels, graded nutrient-specific labels, and single nutrient-specific labels, and Lima et al. [70] studied percentage nutrient-specific labels, graded nutrient-specific labels, and single nutrient-specific labels. Antunez et al. [60] investigated percentage nutrient-specific labels and graded nutrient-specific labels and did not find differences in effects as a function of the labels. Three articles investigated the effects of multiple FOP food labels on digitalized interactive FOP food labels. Two of those three articles found that graded summary labels were most effective in changing the dependent variable than other labels. The remaining article found that single nutrient-specific labels were most effective. Specifically, Finkelstein [48] investigated graded nutrient-specific labels and graded summary labels and found that graded summary labels were the most effective. Blitstein et al. [61] investigated graded summary labels, graded nutrient-specific labels, and combined labels and found that graded summary labels were most effective; and Finkelstein et al. [50] investigated single nutrient-specific labels and graded nutrient-specific labels and found that single nutrient-specific labels were most effective. Regarding digitalized technology-enabled FOP food labels, no article investigated the effects of different types of FOP food labels.

### 4.4. Risk of Bias in Articles

The risk of bias assessment for articles that used a between-participants design with randomization is shown in Appendix A, in Table A5. Out of all 22 articles, 12 articles were assessed as having a high overall risk of bias, nine articles were a moderate risk, and one article was low risk. Regarding the risk of bias domains, a high risk of bias was more common in the deviations from the intended intervention. A moderate risk of bias was more common in reporting of results. Low risk of bias was more common in the missing outcome data domain

The risk of bias assessment for studies that did not use a between-participant design with randomization is shown in Table A6. Six articles had a high overall risk of bias, and three articles had a moderate overall risk of bias. Regarding the risk of bias domain, both high and moderate risk of bias were more common in multiple measurements of outcome pre- and post-intervention domains. At the same time, nine articles had an overall low risk of bias regarding the temporal order of independent variable and effect, the procedure for interventions, measurement of outcome, and reliability of outcome domain.

## 5. Discussion

### 5.1. General Interpretation

This systematic review aimed to present a classification system and investigate the effects of physical and digitalized FOP food labels on healthy food-related behavior. Specifically, the articles that were collected investigated the effects of physical and digitalized static, interactive and technology-enabled FOP food labels on consumer purchases, consumption, hypothetical choices, and self-reports regarding healthy foods. To the best of the authors’ knowledge, this study is the first study to do so.

The results show a difference in the dependent variables defined in this review as a function of digitalized FOP food labels when analyzed individually. Based on the articles included in this review, a higher percentage of articles found a difference in purchase and self-reports regarding digitalized FOP food labels compared to the percentage of articles that used physical FOP food labels. A similar percentage of articles found a difference in hypothetical choices regarding digitalized FOP food labels compared to articles that investigated physical FOP. No articles investigated consumption as a function of digitalized FOP food labels. Hence, the results indicate that the effects of digitalized FOP food labels are greater for purchase and self-reports compared to physical FOP food labels. Furthermore, in the context of digitalized static FOP food labels, more articles reported a change of hypothetical choice compared to articles that investigated self-reports.

When analyzed collectively, the results show a difference in the dependent variables defined in this review as a function of digitalized FOP food labels. The results show that the percentage of articles that found differences between the dependent variables was the same when articles investigated physical and digitalized static FOP food labels. The results also show that the percentage of articles that found a difference between the dependent variables was lower as a function of digitalized interactive FOP food labels than physical FOP food labels. In addition, a higher percentage of articles found a difference in the dependent variable as a function of digitalized technology-enabled FOP food labels compared to physical FOP food labels. Hence, the percentage of articles that found an effect was similar for digitalized static, lower for digitalized interactive, and higher for digitalized technology-enabled FOP food labels than physical FOP food labels when the dependent variables were analyzed collectively. However, more studies are needed to evaluate the effects of digitalized FOP food labels.

Lastly, when compared to different types of FOP food labels and their effectiveness in changing healthy food-related behaviors, combined labels were documented as the most effective for physical, graded nutrient-specific for digitalized static, graded summary labels for digitalized interactive FOP, and no articles investigated the effects digitalized technology-enabled FOP food labels. However, these articles did not compare the same type of FOP food label. Further research is needed to identify which type of FOP food labels are more effective when presented as physical, digitalized static, digitalized interactive, and digitalized technology-enabled FOP food labels.

The articles identified in the introduction and articles which were included in the review had different results in regard to physical labels, to some degree similar results in regard to digitalized static, non-consistent results in relation to digitalized interactive, and was the same article in regard to digitalized technology-enabled FOP food labels. The results of the articles in this review do not align with the results in the literature review. Specifically, one article did not find the difference in purchase [57], one article did find a difference in consumption [75], two articles found a difference in hypothetical choice [51,58], and one did find differences in self-reports article [56] while two articles did not find a difference in self-reports [33,57] as a function of physical FOP food labels. In contrast to previous research, one article did find differences in purchases when self-reports regarding self-control were high [34], one article did not find a difference in hypothetical choice [35], and all three articles found differences in self-reports as a function of physical FOP food labels [19,34,35]. Digitalized static FOP food labels and their effects on healthy food-related behavior based on the articles included in this review are, to some degree, in line with the results of prior research identified in the literature review section. In our systematic review, eight articles did find differences in hypothetical choices [52,53,54,58,59,62,63,73,74], three articles did find difference in self-reports [63,65,71], while two articles did not find difference in self-reports [64,69] as a function of digitalized static FOP food labels. Based on the studies that were identified in the literature review, two articles did find differences in hypothetical choice [17,37], and two articles did find differences in self-reports [36,37] as a function of digitalized static FOP food labels. The effects of digitalized interactive FOP food labels on healthy food-related behavior, based on the articles included in this review, are non-consistent with effects identified prior research mentioned in the literature review section. In our review, two articles found differences in purchases [48,50] while one article did not find differences [67], one article found differences in hypothetical choice [61] while one article did not find differences [52]. Regarding research in the literature review, one article did not find the difference [41] while one article found an increase in purchase [42], two articles found a difference in hypothetical choice [38,39], and one article found a difference in self-reports [40] as a function of digitalized interactive FOP food labels. The effects of digitalized technology-enabled FOP food labels and their effects on healthy food-related behavior were the same as in articles identified in this review and previous research [15] indicating a lack of research regarding digitalized technology-enabled FOP food labels, as shown in Table 1.

There are several alternative explanations regarding the general interpretation of this review. Firstly, this systematic review found different results when healthy food-related behavior was analyzed collectively or individually. When healthy food-related behaviors were analyzed collectively, then similar percentages of articles that found differences in the dependent variable as a function of physical and digitalized static FOP food labels were found. However, when the percentage of articles was analyzed across the dependent variables, a higher percentage of articles were found that documented a change in self-reports as a function of digitalized static FOP food labels compared to physical FOP food labels. One possible explanation is that the search strategy that was used found more articles that investigated the effects of digitalized static FOP food labels on hypothetical choices. The results may have been different if the search strategy identified an equal number of articles that investigated the effects of physical, digitalized static, digitalized interactive, and digitalized technology-enabled FOP food labels on purchase, consumption, hypothetical choice, and self-report. In addition, hypothetical choices regarding preference often involve repeated evaluations by the participants while self-reports may require a single evaluation. This may have impacted the results. Secondly, more articles support that physical and digitalized FOP food labels change hypothetical choices than do articles that used self-reports as the dependent variable. One possibility is that it may be more practical to measure several self-report measurements, such as ratings of healthfulness, taste, affect, and so on, through a questionnaire that presents several such questions compared to measuring several hypothetical choices. The increase in measurements with self-report may increase the probability of not finding changes.

These findings imply that physical and digitalized labels may have different impacts on consumer behavior and there may be several possible mechanisms for these findings. First, it may be the case that the actual sight and symbolic representation of food products may have different impacts on healthy food-related behavior. For instance, Huyghe and collages [76] conducted a series of experiments regarding online and offline grocery shopping on healthy food-related behaviors. Their results indicate that symbolic representation of a product may impact self-control and that may impact healthy food purchases [34]. Furthermore, online grocery stores provide the possibility of presenting pre-selected food products along with recipes (meal kits). The effects of digitalized FOP food labels may function differently when they are based on a collection of many products compared to individual products. Finally, consumers can use a variety of sensory modalities to assess a food product before purchasing it in a physical store whereas online grocery stores provide no specific sensory information such as smell and touch. In an online grocery store, one can present textual descriptions of sensory information based on the association between previously purchased products; for instance, a message at the point of purchase that suggests that a particular brand of apple has the same “taste profile” as other previously purchased products. The presence or absence of these variables may influence the effectiveness of certain digitalized FOP food labels.

### 5.2. Limitation of Evidence Based on Articles and Review

There exist several limitations of evidence based on the articles. Firstly, the majority of the articles included in this review had an overall moderate or high risk of bias. Specifically, reporting of results was the risk of bias domain that had the least “Low risk” assessments. The majority of these articles had a moderate risk of bias, where there was no provided information on a pre-specified statistical analysis plan. Secondly, the majority of the articles investigated the effects of digitalized static FOP food on hypothetical choices or self-reports. There is a lack of articles examining the effects of digitalized static FOP food labels effects on purchase and consumption, the effects of digitalized interactive FOP food labels on consumption and self-reports, and the effects of digitalized technology-enabled FOP food labels in general, except for one article.

Several methodological limitations of this review are also worth addressing. Firstly, the search strategy may have been too restrictive regarding identifying all articles investigating the effects of physical and digitalized FOP food labels. However, this systematic review aimed to investigate the effects of physical and digitalized FOP food labels on healthy food-related behavior and not to identify all previous research that has examined FOP food labels. Furthermore, this systematic review used only principal search systems which do not include Google Scholar. This may have impacted the number of articles that have investigated the effects of digitalized static FOP food labels included in the review. However, the number of articles that investigated interactive and technology-enabled FOP food labels produced by search engines is not likely to be affected. Secondly, one reviewer did the risk of bias assessment and the data collection process, apart from the findings of articles that another independent reviewer assessed. Thirdly, tools regarding the certainty of assessment were not used. However, issues regarding the articles identified were discussed in regard to limitations of evidence based on the studies. Several articles included in this review had an overall moderate to high risk of bias. Hence, presenting a meta-analysis that investigated the degree of effects was not appropriate. Furthermore, several articles used different measurements concerning the dependent variable. For instance, one article could present three food products, and another could present six food products when measuring hypothetical choices regarding healthy food. A meta-analysis would be appropriate if the articles in this review used the same experimental paradigm. However, a meta-analysis was not done due to a large variation in experimental designs. This review instead investigated how many empirical articles found a difference in healthy food-related behavior as a function of physical and digitalized FOP food labels instead of directly comparing different dependent variables. Lastly, this review did not control for the confounding effects of nutritional fact labels. Future studies could address this and examine the confounding effects of nutritional information labels and digitalized FOP food labels. To control for the confounding effects would require a larger sample of empirical studies that met the inclusion parameters of this study. However, in this study, it was not feasible to perform such an analysis. Future studies could address this and examine the confounding effects of nutritional fact format and digital labeling format.

### 5.3. Implications and Further Research

Several implications exist based on the findings of this review. Firstly, this review found an increase in research articles regarding digitalized FOP food labels. This trend indicates that investigations into the effects of such labels are increasing and will be important for future online grocery retailing practices. Secondly, this review found one article [15] which investigated the effects of digitalized technology-enabled FOP food labels and found a reliable increase in healthy food purchases and a decrease in unhealthy nutrients while at the same time not showing significant differences in dollars spent per kcal, indicating that one indeed can increase healthy food purchase without decreasing profit. As mentioned, digitalized technology-enabled FOP food labels may increase the number of healthy food purchases, attract new customers, and increase the positive reputation of online grocery stores and brand owners such as Thrive Market, Tesco, Sainsbury, Walmart, and AmazonFresh. Even though digitalized interactive FOP food labels alone may not change purchases, such labels may still provide consumers with accurate descriptions of food products. They may attract new consumers and increase positive word of mouth regarding brand owners and retailers.

One way to advance further studies regarding digitalized technology-enabled FOP food labels is to conduct controlled laboratory experiments regarding how previous neutral symbols or stimuli may acquire a function as a healthy food label in individual analysis. As mentioned, the effects of such labels may increase the purchase of healthy foods for some subgroups [5]. Previous history with interactions with these labels may be one variable that may impact their effectiveness. The majority of the articles studied healthy food labels implemented through public policy. Participants presumably had some history regarding such labels and this may influence the effectiveness of such labels in changing healthy food-related behavior. Although further studies regarding how different FOP food labels impact different subgroups are needed [7], there is also a need to identify how the presentation of information or stimuli, in general, may impact healthy food-related behavior on an individual psychological and behavioral level, and then replicate such findings on a large scale level. Future studies regarding digitalized technology-enabled FOP food labels could, in combination, investigate the effects of automatic self-monitoring of healthy food purchases, presentation of healthy food labels on food products that have not been purchased previously in order to increase variability regarding food choices, decreasing the delay of future consequences by presenting real-time based health-related information such as a decreasing in the chance of illnesses associated with healthy food purchases, self-imposed costs or restriction of unhealthy foods in the combination of single nutrient-specific labels, other consumers FOP food label scores, and immediate or delayed presentations of such labels (e.g., real-time based versus every third purchase based on products in virtual basket). One way to advance further studies regarding digitalized interactive FOP food labels is to examine their effects on hypothetical choice, self-reports, and food consumption. As mentioned, articles that have used self-reports as the dependent variables may have measured various constructs such as healthfulness, trust, familiarity, etc., compared to hypothetical choice. Further studies could investigate such constructs.

## 6. Conclusions

In conclusion, digitalized interactive and technology-enabled FOP food labels and their effects on healthy food-related behavior remain an unexplored research area. This systematic review identified previous research regarding physical, digitalized static, digitalized interactive, and digitalized technology-enabled FOP food labels and investigated their effects on healthy food-related behavior regarding purchase, consumption, hypothetical choice, and self-reports. When analyzed collectively, a similar percentage of articles demonstrated the effects of physical and digitalized static FOP food labels on healthy food-related behavior. Furthermore, a lower percentage of articles demonstrated the effects of digitalized interactive FOP food labels compared to physical FOP food labels. However, a higher percentage of articles demonstrated the effects of digitalized technology-enabled FOP food labels. When analyzed individually, a higher percentage of articles supported a difference in purchase and self-reports as a function of physical and digitalized FOP food labels. Regarding articles identified in this review that compared different types of FOP food labels, including physical combined, digitalized static graded-nutrient, and digitalized interactive graded summary, FOP food labels were most effective. Our results show that there is an increase in the publication of studies regarding digitalized FOP food labels and their effect on healthy food-related behavior. Knowledge regarding variables that moderate these effects would be important for future studies.

## Figures and Tables

**Figure 1 behavsci-12-00363-f001:**
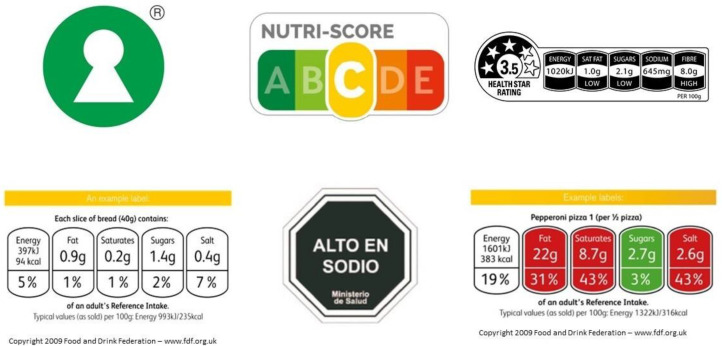
The figure shows examples of different types of front-of-package food labels. From top left to right, a single summary label (Nordic Keyhole), graded summary label (French Nutri-score), and combined label (Australian Health Star Ratings) are shown. A percentage-based nutrient-specific label (British Guideline Daily Amounts), single nutrient-specific label, and graded nutrient-specific label (British Traffic Lights) are shown from bottom left to right.

**Figure 2 behavsci-12-00363-f002:**
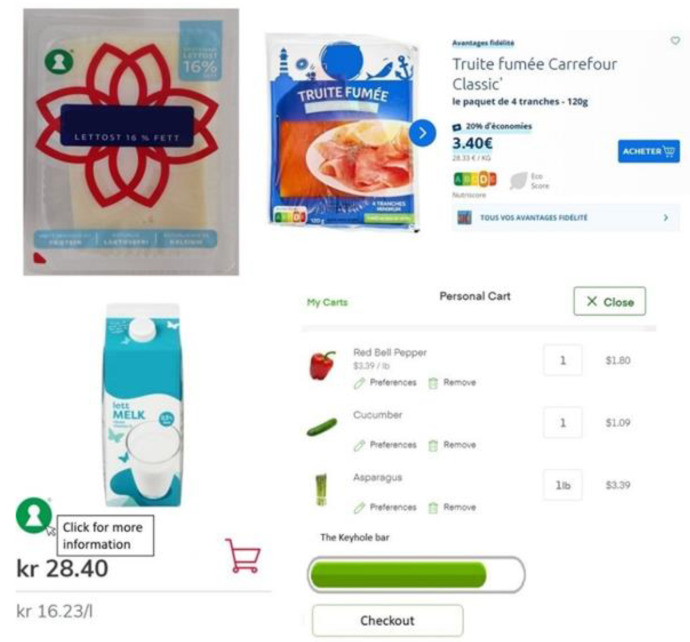
The figure shows hypothetical examples of physical (upper left), digitalized static (upper right), digitalized interactive (lower left), and digitalized technology-enabled (lower right) front-of-package food labels.

**Figure 3 behavsci-12-00363-f003:**
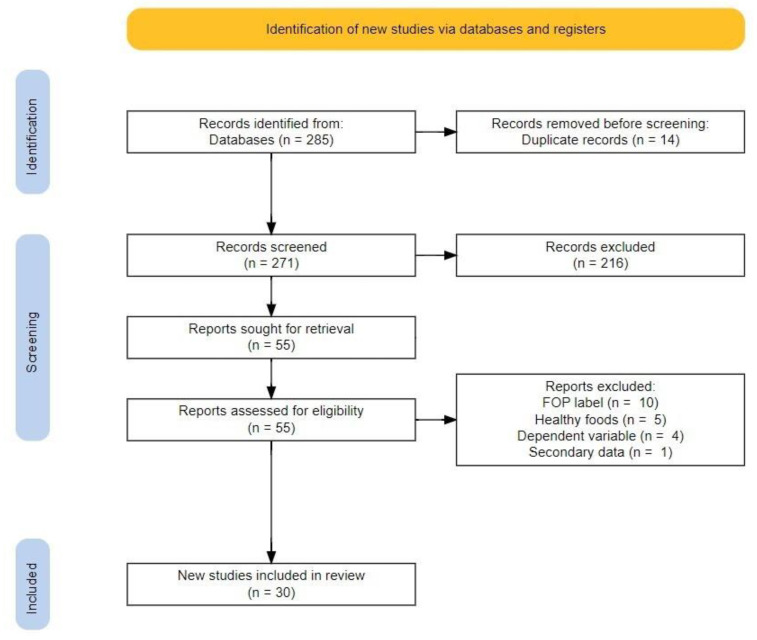
The flowchart shows the identification, screening, and inclusion of records, reports, and articles included on the left and reasons for removal on the right.

**Figure 4 behavsci-12-00363-f004:**
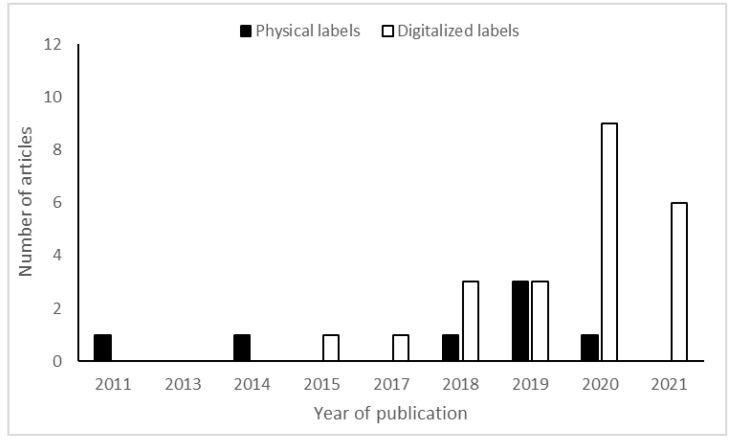
The *Y*-axis shows the number of articles included in this review, and the *X*-axis shows the year of publication for its corresponding article. The black bar corresponds to articles that investigated physical FOP food labels while the white bar corresponds to articles that investigated digitalized FOP food labels.

**Table 1 behavsci-12-00363-t001:** The effects of physical and digitalized FOP food labels across dependent variables.

	Physical	Digitalized		
Dependent Variable		Static	Interactive	Technology-Enabled
Purchase	0% (10)		66.67% (1, 3, 20)	100% (25)
Consumption	100% (30)			
Hypothetical choice	100% (4, 11)	100% (6, 7, 11, 12, 15, 16, 28, 29)	50% (5, 14)	
Self-reports	33% (9, 10, 26)	60% (16, 17, 18, 22, 24)		

*Note*. The table shows the percentage of articles that indicate that the corresponding dependent variable was under full control of physical, digitalized static, digitalized interactive, and digitalized technology-enabled FOP food labels based on articles that had a presence and an absence of FOP food label conditions. Each article’s number in the parentheses specifies the articles.

## Data Availability

All data is provided in this article or in the Appendix A.

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
