# Peer review of "Effects of Digitalized Front-of-Package Food Labels on Healthy Food-Related Behavior: A Systematic Review"

_behavsci, 2022, doi:10.3390/bs12100363_

Round 1

Author Response

See the attached PDF file.

Reviewer 2 Report

The writing is fairly good and easy to read. The only thing that is not good is how the author organizes the literatures in each table. Too many tables look like literatures parade.The conclusion is fairly ok. Actually, more words are better for the current conclusion.

The main picture is the effects of FOP food labels on healthy food-related behavior in relation to which type of behavior is measured. The author then concludes that no systematic literature review has examined the effects of physical and digitalized FOP food labels regarding to different types of labels and their effect on healthy food-related behavior. Knowledge of such effects on healthy food-related behavior has both academic and social importance with regards to the understanding of how technology in this setting influences human behavior, health-related behavior, and consumer behavior, while at the same time developing knowledge that may aid reducing obesity, economic costs, and human suffering worldwide. In addition, knowledge regarding digitalized FOP food labels may give brand owners and retailers a competitive advantage. Specifically, digitalized interactive and technology-enabled FOP food labels may provide more accurate descriptions regarding products and present personalized, dynamic, and real-time information of health-scores to consumers which may increase the value of healthy foods while at the same time benefit brand owners and retailers by increasing number of healthy food purchases, attracting new customers, and increase positive word of mouth.

The topic of FOP food labels is good and relatively relevant. This paper is able to provide some more insight on the labeling and its related behaviour.

Please categorise the content in Table S1 based on Methodological approach.

Please categorise the content in Table S2 based on food labels.

Author Response

See the attached PDF file.

Reviewer 3 Report

Title should not be started with “The”.

Topic is interesting its review article should be focused as compare to systematic review as there are some studies showing better results.

Revise the sentence  in abstract “This review used “Web of Science,” “Science Direct,” “PubMed,” 11
and “Wiley Online Library,” to collect articles that investigated the”….. you can write it Data collected fir this review or something else but this should be change.

Add results/findings in abstract.

Add a conclusive line in abstract as well and  future aspects of this review should not be mentioned in the abstract. You can write the last line of abstract in recommendation section.

Introduction should must be revised there are some irrelevant material add specifically about your topic.

Change these sentence grammatically these are not correct one.  First, previous 113
research regarding the classification of FOP food labels is provided. Secondly, previous 114
research is presented regarding the physical FOPs food labels, digitalized static, interac- 115
tive, and technology-enabled FOP food labels. Thirdly, the methodological approach of 116
this review is provided. Fourthly, the result of the review is presented. At last, the final 117
part presents a discussion and further research directions. 11”..

Authors can add some more relevant and updated studies in review of literature section.

Author should must add inclusion and exclusion of articles criteria.

Author Response

See the attached PDF file.

Round 2

Reviewer 1 Report

I believe the authors have address my comments adequately.

Author Response

We would like to thank you for your helpful comments.

Reviewer 3 Report

Authors have improved the article now it can be considered for publication

Author Response

(The authors gave the same response as above.)
